# Optimization of Hazelnut Spread Based on Total or Partial Substitution of Palm Oil

**DOI:** 10.3390/foods12163122

**Published:** 2023-08-20

**Authors:** Francesco Marra, Arianna Lavorgna, Loredana Incarnato, Francesca Malvano, Donatella Albanese

**Affiliations:** Department of Industrial Engineering, University of Salerno, 84084 Fisciano, Italy; fmarra@unisa.it (F.M.); arianna.lavorgna.95@gmail.com (A.L.); lincarnato@unisa.it (L.I.); dalbanese@unisa.it (D.A.)

**Keywords:** hazelnut spread, oleogel, design of experiment

## Abstract

Palm oil is widely used in the manufacturing of hazelnut-based spreads due to its unique fatty acid and triacylglycerol profile and, thus, its crystallization behaviour, which makes it suitable for use in fat-based spreadable products. An interesting solution that enables the replacement of palm oil is given by oleogels made with high nutritional quality oil. In this study, the influence of the replacement of palm oil with different glycerol monostearate/olive oil-based oleogels, as well as the influence of the different amounts of GMS employed in oleogel preparation, on the oil binding capacity, spreadability, and rheological and sensory parameters of hazelnut cocoa spreads was investigated. A design of experiment (DoE) approach, with the adoption of the D-optimal design, was used to plan the cocoa hazelnut spread formulations, with the aim being to identify the optimal formulation with desirable quality parameters in terms of Casson’s viscosity, spreadability, and oil binding capacity. The resulting optimized formulation was identified in a spread characterized by a total replacement of palm oil with an oleogel made of 95% olive oil and 5% GMS.

## 1. Introduction

Anhydrous cocoa hazelnut cream is a complex spreading system consisting of solid particles dispersed in a continuous fluid [1]. The main characteristics affecting the quality of this product, like cream stability, spreadability, and melting properties, principally depend on the type of fat used in it. Palm oil is widely used in the manufacturing of confectionary and hazelnut-based spreads due to its unique fatty acid (FA) and triacylglycerol (TAG) profile and, thus, its crystallization behaviour, which makes it suitable for use in fat-based spreadable products [2]. Despite its technological properties, palm oil is affected by two main issues related to risks to human health and environmental impact [3]. Health risks are related to the monochloropropanediol ester (MCPDE) and glycidyl ester (GE) that are undesirably produced from mono- and di-acylglycerol during the refining process, especially during the deodorization step [4]. Regarding environmental concerns, the increase in palm oil plantations replaces ancient forests with monocultures that cannot support the same levels of biodiversity.

Previous studies have been carried out regarding the substitution of palm oil with other edible vegetable oil.

In cocoa hazelnut spread manufacturing, palm oil has been substituted with coconut and milk fat [5]. In both cases, hazelnut spread showed lower values in every texture attribute and viscosity than the values observed in palm oil-based spreads.

Aydemir et al. [6] studied the possibility of using different fats (cocoa butter, hydrogenated palm oil, and margarine) and oils (hazelnut oil, anhydrous milk fat, sunflower seed oil, olive oil, and coconut oil) in the production of cocoa hazelnut spreads. Among the experimental samples, spreads with hazelnut, margarine, and sunflower seed oils showed the highest flavour and aroma scores. As fat or oil solidity increased, the oil separations and the flow behaviour index decreased; however, the spreadability, stickiness, and viscosity increased.

An interesting solution employed for the replacement of palm oil is the use of oleogels made using oil of high nutritional quality [7,8,9,10,11,12].

Oleogelation is a promising and valid technique used to convert liquid vegetable oil into a solid-like gel (oleogel) with similar textural and thermal features, as well as rheological properties, to solid-like fats. Oleogels can be defined as semisolid systems characterized by a continuous phase of a hydrophobic liquid (such as vegetable oil), in which a three-dimensional network (composed of an oleogelator) is responsible for the physical entrapment of the liquid [13,14].

Oleogels are classified based on the type of structurants (fatty acid derivates, sterols, and polymers) that are responsible for different gelation mechanisms. Among various lipid-based gelators, natural waxes and saturated monoglycerides are widely studied with the purpose of replacing saturated and trans fats in different food products, such as meat patties [15], ice cream [16], cream cheese [17], confectionary filling [18], and cakes [7].

To date, very few studies exist regarding the use of oleogels in chocolate spreads. Bascuas et al. [19] and David et al. [20] investigated the influence of biopolymer (HPMC and cellulose derivates, respectively) in the preparation of oleogel as a fat in chocolate spread. Fayaz et al. [21] studied the influence of saturated monoglyceride and wax-based oleogel-palm oil mixture on the structural and rheological properties of chocolate spreads. Borriello et al. [22] evaluated the rheological properties and physical stability of hazelnut cocoa creams prepared using pumpkin seed oil and beeswax-based oleogel. To the best of our knowledge, no studies have been carried out regarding the influence of GMS-based oleogel as a partial or total fat replacement on the quality parameters of cocoa hazelnut spread. Based on the above information, this study aimed to investigate the influence of different replacements (0–100%) of palm oil with glycerol monostearate (GMS)/olive oil-based oleogels on the oil binding capacity, spreadability, and rheological and sensory parameters of hazelnut cocoa spreads. Moreover, the influence of the different amounts of GMS employed in oleogel preparation on the quality parameters of spread samples was evaluated.

A design of experiment (DoE) approach was used to plan the cocoa hazelnut spread formulations to identify the optimal formulation with desirable quality parameters in terms of Casson’s viscosity, spreadability, and oil binding capacity.

## 2. Materials and Methods

### 2.1. Materials

The basic ingredients used in cocoa hazelnut spread formulation, such as sugar (sucrose), hazelnut paste, olive oil, palm oil, cocoa powder, whey proteins, milk powder, soy lecithin, and vanillin, were bought at a local market. Crude palm oil (PO) (melting point 32.34 ± 0.10 °C) was purchased from Naissance (Neath, UK). Glycerol Monostearate (GMS) was obtained from Axenic Health Solutions (Plano, TX, USA).

### 2.2. Oleogel Preparation

Oleogels were prepared according to the method of Malvano et al. [7], with some modifications, using olive oil (component 1 of mass composition x_1_) and GMS (component 2 of mass composition x_2_). Different amounts of GMS, ranging between 3 and 6% (*w*/*w*), were dissolved in olive oil (OO) at a temperature above the melting point of GMS (>70 °C) under magnetic stirring at 100 rpm. After the complete oleogelator dissolution, the mixtures were placed in a refrigerated bath under static conditions at 5 °C for 30 min until the samples reached room temperature (25 ± 2 °C), recording an average cooling rate of 2 °C/min. The prepared oleogels were stored at room temperature for at least 24 h before being analyzed.

### 2.3. Preparation and Characterization of Cocoa Hazelnut Spreads

The cocoa hazelnut spreads formulation included sugar (43%), fat (palm oil or oleogel) (19.5%), hazelnut paste (14.6%), cocoa powder (6%), whey powder (3%), skimmed milk powder (13.1%), soy lecithin (0.7%), and vanillin (0.1%). A food processor (TM21 Thermomix, Vorwrek, Wuppertal, Germany) was used to mix the ingredients. The fat phase (hazelnut paste and palm oil/oleogel) was heated at 55 °C until it reached a homogenous mixture, and the dry ingredients were dispersed in the molten fat phase and blended for 1.5 h. Afterwards, the mixture was refined using a wet grinder (Premier, Chennai, India) for 3 h. The spreads were stored in a cool and dry condition for 48 h before being analyzed.

#### 2.3.1. Rheological Properties

The viscosity of the hazelnut spread samples was measured using a Rheometric Scientific rotational rheometer, ARES (Advanced Scientific Expansion System). The measurements of stress and apparent viscosity were made at shear rates between 0.1 and 100 s^−1^. The tests were conducted at a steady state at 25 °C. Each sample was analyzed in triplicate. To assess the rheological behaviour of the samples, Casson’s model and the Power Law model were considered as constitutive equations, as reported below, as Equations (1) and (2), respectively:(1)$$\sqrt{\tau }=\sqrt{\tau_{ca}}+\sqrt{\eta_{ca}}\;\sqrt{\dot{\gamma }}$$
(2)$$\tau =K{\dot{\gamma }}^{n}$$
where τ is shear stress (Pa), τ_ca_ is Casson’s yield stress (Pa), η_ca_ is Casson’s plastic viscosity (Pa s), K is the flow consistency index (Pa s^n^), n is the flow behaviour index (-), and $\dot{\gamma }$ is the shear rate (s^−1^).

#### 2.3.2. Oil Binding Capacity

The oil binding capacity (OBC) of coca hazelnut spreads was evaluated according to the method of Aydemir et al. [6] with some modifications. About 8 g of sample was placed into a tube and centrifuged for 15 min at 5000 rpm. Oil separated after centrifugation was pipetted out, and the weight of the remaining spread was determined. OBC was calculated as follows:(3)$$OBC\;\left(\%\right)=\left[1-\frac{\left(m_{0}-m_{}\right)}{m_{0}}\right]\;100$$
where *m*_0_ is the mass of the initial spread, and *m* is the mass of the spread after centrifugation. Three replicates were performed for each sample.

#### 2.3.3. Texture Measurements

A texture analyzer (LRX Plus, Lloyd Instruments, Chicago, IL, USA) was employed to investigate the textural properties (spreadability and hardness) of hazelnut spread samples. The measurements were performed according to Kangchai et al. [23], with a TTC Spreadability Rig and a 90° cone probe at room temperature; a sample of spread was put into the female cone and penetrated (18 mm) using the corresponding conical probe at a speed of 30 mm/min. The maximum force used to penetrate the upper probe in the sample represents the hardness (in N), and the spreadability was measured as the area (in N s) of the compression test (Force vs. time). Smaller values of the area reflect greater spreadability. All measurements were performed in triplicate.

#### 2.3.4. Sensory Analysis

Sensory analysis of spreads was carried out by a panel made of 9 persons trained in the study of cocoa hazelnut spreads according to Albanese et al.’s method [24].

The judges were chosen from members of the Department of Industrial Engineering of the University of Salerno, using as the selection criteria time availability and no nut allergy/intolerance. The pre-selected judges were selected based on their ability to identify odors and the 5 basic tastes, and we used ISO 3972 [25] and ISO 5496 [26] to recruit 9 assessors. Triangle tests were used as methods to perform the evaluation of the performance of the judges [27]

The descriptors chosen to perform the sensory analysis of spread samples, as well as their evaluation method, are reported in Table 1. As a reference standard, we used a market leader cocoa hazelnut spread.

The samples were served, at room temperature, in white porcelain cups coded with three digits to judges along with toasted bread slices and plastic spoons. The sample evaluation order was randomized according to the incomplete block design. The judges rated the samples using a 9-point scale; firstly, the fluidness and spreadability were measured as texture attributes, and the meltability, adhesiveness to mouth, and flavour were used as taste attributes. For the spreadability, the judges were asked to spread the sample on the toasted bread slice. The assessors evaluated the control spread and the 15 different spreads under study in different sensory analysis sessions.

All procedures performed in this study involving human participants were carried out in accordance with the 1964 Helsinki Declaration and its later amendments.

### 2.4. Experimental Design and Data Analysis

In this study, the design of experiment (DoE) method was employed to investigate the effect of replacing the fat contribution in the spread formulation (given by the palm oil) with olive oil-based oleogels on the main quality parameters of cocoa hazelnut spreads. Thus, the control cocoa hazelnut spread was modified by replacing the palm oil, totally or partially, with olive oil-based oleogel. This process led to us defining the fat mixture of oleogel and palm oil as a binary system of compositions z_1_ and z_2_, respectively. The influence of oleogel composition on the investigated properties of the spread formulation was also studied. In particular, a D-Optimal design was employed, and the constraints of ingredients (independent variables) were established as follows: the percentage of olive oil in the oleogel formulation (x_1_, ranging from 94% to 97%), percentage of GMS in the oleogel formulation (x_2_, ranging from 3% to 6%), and fractions of palm oil (z_2_) at 0%, 25%, 50%, and 75%. Table 2 shows the experimental design, including 15 different formulations. The responses (dependent variables) evaluated for each formulation were Casson’s viscosity (y_1_), spreadability (y_2_), and oil binding capacity (y_3_).

JMP Statistical Software (SAS Institute. Inc., Cary, NC, USA) was used to organize the experimental design, analyze the experimental data, and identify the optimal cocoa hazelnut spread formulation.

An empirical first-order polynomial model was employed to model the experiment outputs in terms of Casson’s viscosity, spreadability, and oil binding capacity as a function of reduced oleogel compositions.
(4)$$y_{i\;}={\beta_{i}}_{1}x_{1}^{*}+{\beta_{i}}_{2}x_{2}^{*}+\beta_{i12}x_{1}^{*}x_{2}^{*}+\beta_{i13}x_{1}^{*}+\beta_{i23}x_{2}^{*}+\beta_{i123}x_{1}^{*}x_{2}^{*}\;\left(x_{1}^{*}{-x}_{2}^{*}\right)$$
where *y_i_* is the studied response (i.e., Casson’s viscosity, spreadability, and OBC), $x_{1}^{*}$ and $x_{2}^{*}$ are the reduced oleogel compositions, defined as $x_{1}^{*}=\frac{x_{1}-x_{1min}}{x_{1max}-x_{1min}}$, $x_{2}^{*}=\frac{x_{2}-x_{2min}}{x_{2max}-x_{2min}}$. $\beta_{i1}$, $\beta_{i2}$, $\beta_{i12}$, and $\beta_{i123}$ are the pure regression parameters, while $\beta_{i13}$, and $\beta_{i23}$ are the regression parameters, depending on the mass fraction of palm oil in the fat mixture (z_2_).

The coefficient parameters of the developed models were estimated via multiple linear regression analysis based on the least-squares method and statistically evaluated via analysis of variance (ANOVA). The best-fitted model was selected via the evaluation of the statistical parameters, such as the coefficient of determination (R^2^), adjusted coefficient of determination (adjusted R^2^), standard deviation, and lack-of-fit and regression data (*p*-value and F value) [28].

The desirability function analysis (DFA) is one of the methods widely used in industry to deal with the optimization of multiple response processes. The DFA transforms the response to a desirability function that takes values in the range 0 < d < 1: desirability will be 1 if the response variable is at its goal, and it will become 0 if the response variable is outside of the acceptable range.

In this study, the desirability function corresponded to an optimal formulation set by minimizing Casson’s viscosity and spreadability while maximizing the oil binding capacity in a range of values that referred to a market leader cocoa hazelnut spread.

All of the analyses were performed in triplicate. Experimental data were reported as mean and standard deviation and subjected to one-way analyses of variance (one-way ANOVA). The significant differences (*p* < 0.05) among the samples were determined via Duncan’s test.

Principal component analysis was used to study attribute–sample relationships. Using PCA, we created a sensory space, in which samples were positioned in the attribute–sample space according to their characteristic sensory attributes [24].

Statistical analyses were performed via the JMP Statistical Software (SAS Institute. Inc., Cary, NC, USA).

## 3. Results

The experimental design with independent variables and the respective experimental responses are shown in Table 3, which we employed to find the optimum cocoa hazelnut spread formulation according to Equation (4).

As reported in Table 4, the model functions were found to have a coefficient of determination value higher than 0.9, showing the high significance of the model, which indicates an excellent agreement between predicted and observed responses. Moreover, the lack of fit was insignificant, having a lower possibility of error.

The effects of the ingredients on the Casson’s viscosity, spreadability, and oil binding capacity of cocoa hazelnut spreads and their interactions were investigated through the study of the estimated parameters of the prediction models for each response (Table 5). When the independent variable significantly (*p* < 0.05) affects the response, the value is listed in bold. A positive term in the regression equation represents an effect that favours optimization due to synergistic effects, whereas a negative term reveals an antagonistic effect between the factors and the response [29].

Based on the developed models, the results showed that the composition of oleogel based on olive oil and GMS assumes an important role in hazelnut spread formulation, significantly (*p* < 0.05) influencing all analyzed responses.

Moreover, Casson’s viscosity and spreadability were influenced by the interaction between GMS and palm oil at 75%, while OBC was influenced by olive oil when palm oil was totally replaced.

### 3.1. Rheological Properties

The rheological parameters evaluated regarding the different cocoa hazelnut spreads were apparent viscosity and yield stress. The latter parameter represents the minimum pressure applied to determine the flow of the fluid. The apparent viscosity is an index of the extent to which the nature of the fluid determines the dissipation of the energy of movement supplied; therefore, it is the ease of movement of the fluid with the same applied stress [1]. As reported elsewhere [30] the rheological properties of spreadable spreads are mainly influenced not only by the fat fraction, but also by the particle size distribution. For this latter issue, no significant differences were found regarding the particle size distribution for all of the spread samples, which ranged from 26.30 ± 0.35 to 33.48 ± 0.76 μm.

The apparent viscosity vs. the shear rate of the control sample and the spread formulation containing the oleogels are shown in Figure 1. All of the tested formulations showed non-Newtonian behaviour, tending to pseudoplasticity since the apparent viscosity decreased with the increase in the shear rate. At low values of the shear rate, the control sample exhibited an apparent viscosity of even seven times higher than those measured for spread formulations based on oleogels, highlighting the role played by palm oil composition in determining the apparent viscosity of the formulated spread. Oleogels at low concentrations of GMS (x_2_ = 3% and x_2_ = 3.7%, in Figure 1a,b, respectively) exhibited apparent viscosity values lower than those of the control along the entire investigated range of shear rate. For oleogels with x_2_ = 4.5% and z_2_ = 75%, the value of the shear rate (2.51 1/s) above which the apparent viscosity of the spread formulation with oleogels was higher than that of the control (Figure 1c). At a low shear rate, increasing the GMS composition in the oleogel brings an increase in the apparent viscosity.

This analysis further justifies the choice of using Casson’s viscosity as the objective function in the response model, as it has a rheological parameter independent of the shear rate. Casson’s estimated parameters were reported in Table 6. Given a certain composition of GMS in the oleogel, Casson’s viscosity increases as the palm oil content in the fat phase increases, indicating that a higher force is necessary to enable sample flowing because of the high-strength structure of the fat phase. Without palm oil being present in the fat phase, Casson’s viscosity significantly increases when the composition of GMS in the oleogel increases. With palm oil composition in the fat phase being 25%, 50%, or 75%, the role of GMS composition in determining the apparent viscosity of the spread cannot be generalized. For palm oil composition of 25%, the apparent viscosity significantly increases when the GSM percentage in the oleogel changes from 3 to 4.5%. Further increasing in GSM composition does not lead to any appreciable increase in the apparent viscosity.

Good rheological properties in chocolate spreads made using a mixture of palm fat and monoglyceride-based oleogels were found by Fayaz et al. [21]. The authors attributed their results to chemical compatibility between the palm oil and the oleogel, as well as a strengthening of the structure because of the formation of lamellar structures stabilized by hydrogen bonds.

### 3.2. Oil Binding Capacity

Oil binding capacity (OBC) is an important parameter that provides information about the measure of the oil migration, which is a frequent phenomenon in spreads and could be perceived as a quality defect by consumers.

The measured OBC values (Table 7) indicate that all samples had excellent oil binding properties since oil release upon centrifugation was lower than 7%. The lowest OBC values were recorded in spreads characterized by the total replacement of palm oil with oleogel and low amounts of oleogelator (3%, 3.7% and 5.4% of GMS), while spreads made with partial palm oil replacement (50% and 75%) presented OBC values similar to those of the control spread. These results highlighted the high capability of developed oleogel-based hazelnut spreads to keep oil, as well as their high physical stability. According to Lopez-Martinez et al. [31], the capability of GMS to gel vegetable oils is associated with the formation of inverse lamellar phases, which are stabilized by strong hydrogen bonds.

When the palm oil was totally replaced, its OBC was close to 93%, compared to the 99% OBC exhibited by the control sample, thus confirming the good ability to retain oil shown by the oleogel.

### 3.3. Texture Properties

Hardness and spreadability are the principal attributes that influence the perceived quality among consumers of cocoa-based anhydrous creams. The hardness and spreadability of the different cocoa hazelnut spreads were reported in Table 8. Our results were in agreement with those of several other authors [23,31,32] who reported that hardness is significantly associated with spreadability, i.e., the higher the hardness, the greater the resistance to spreading. For the same GMS amount in the oleogel, the hardness, as well as the spreadability, significantly (*p* < 0.05) decreased in line with decreasing palm oil percentage in the formulations. The lowest values were recorded for spreads formulated based on the total replacement of palm oil with oleogel. The results agreed with Fayaz et al. [21], who asserted that the firmness of chocolate spreads is positively correlated with the firmness of the solid fat phase, as they found that the partial substitution of palm oil with pomegranate seed oil oleogels based on monoglycerides oleogels reduced the firmness of chocolate.

Furthermore, by comparing spreads made using the same amount of oleogel but different percentages of GMS, no significant differences were observed, except for the GMS _6PO_75 sample, which showed higher hardness and spreadability values than those of the control. The texture results are consistent with those of the rheology, for which the spread samples with higher palm oil and GMS contents showed more noticeable viscoelastic behaviour.

### 3.4. Sensory Evaluation

The relationships between the sensory attributes and the different hazelnut spread samples investigated in this study were performed through principal component analysis (PCA). Bidimensional representations of PC1 and PC2 scores for attributes and samples are shown in Figure 2. The main differences and similarities between the attributes and each sample were explained based on the first and second principal components (PC), which explained 74.7% and 14.9% of the total variance, respectively. PC1 separates, in a clear way, the hazelnut spreads with the highest contents of palm oil set by the experimental plan (50% and 75% PO), as well as those with the highest contents of GMS in the oleogel (6%), from all of the other samples, in particular the hazelnut spreads made using 100% oleogel. Regarding the influence of sensory attributes on the investigated hazelnut spreads, it has been possible to observe that for all of the samples with high amounts of palm oil (75% of the fat phase used), the control (100% palm oil) and those with a simultaneous presence of high GMS and palm oil are characterized by adhesiveness to the mouth, that is, negatively correlated with spreadability, meltability, and fluidness parameters.

This result could be explained based on the high melting temperature of pure palm oil and olive oil GMS oleogel, according to our previous results, which estimate the melting temperatures of palm oil and oleogel GMS 6% to be close to 32.34 °C and 51.53 °C, respectively.

Moreover, as reported elsewhere, GMS seems to influence the crystallization of palm oil and vegetable oil blends, increasing the strengthening of the fat crystalline network and, thus, the melting point [21]. On the contrary, the spread samples prepared using only GMS-based oleogels and those with the lowest percentages of palm oil were highly correlated with the most important technological attributes of spread-based food products, such as spreadability, fluidness, and meltability, which highlights the proper melting properties of the samples in the mouth. No correlations have been found between flavour and the other sensory attributes investigated.

### 3.5. Multivariable Optimization

The mathematical models reached in this study have been used to predict the optimum spread formulation. Multiple optimizations were performed to identify the optimal amount of olive oil and GMS, as well as the palm oil replacement, to obtain a spread with the desired properties in terms of Casson’s viscosity, spreadability, and oil binding capacity, as explained in Section 2.4.

The resulting optimized formulation was identified in a spread characterized by the total replacement of palm oil with an oleogel made of 95% olive oil and 5% GMS. This ingredient combination achieved an overall desirability score of 0.9989, indicating that the quality of the product was acceptable and excellent [33].

## 4. Conclusions

This work investigated the use of D-optimal design for the replacement of palm oil with olive oil-based oleogel in cocoa hazelnut spread formulation to identify the best formulation that had the desired quality parameters. Using the design of experiment technique, 15 different spread formulations were prepared and analyzed in terms of their oil binding capacity, Casson’s viscosity, and spreadability, which represented the set of desired properties. This approach led to the determination of a mathematical model for each of the above-mentioned properties, and we then used these models to calculate the optimal ingredient combination to reach the highest overall desirability. The statistical analysis of the results highlighted that the spreadability, as well as the hardness, significantly decreased with the decrease in the palm oil percentage in the formulations, given a certain amount of glycerol monostearate in the oleogel. The lowest values were recorded for spreads formulated via the total replacement of palm oil with oleogel. The texture results were consistent with those of the rheology, with the spread samples with higher palm oil and glycerol monostearate content showing more noticeable viscoelastic behaviour. Finally, the oil binding capacity results indicate that all samples had an excellent ability to retain oil since oil release upon centrifugation was lower than 7%. The results of the optimization revealed that the best mixture was the formulation characterized by a total replacement of palm oil with an oleogel made of 95% olive oil and 5% glycerol monostearate, according to the desirability function approach.

The overall result of this study highlighted the potential use of the design of experiment technique, together with the D-optimal design, to formulate food spreads, particularly when oleogels are used to replace saturated fats in food formulations, which require the employment of solid fats to give them specific technological and sensory characteristics.

## Figures and Tables

**Figure 1 foods-12-03122-f001:**
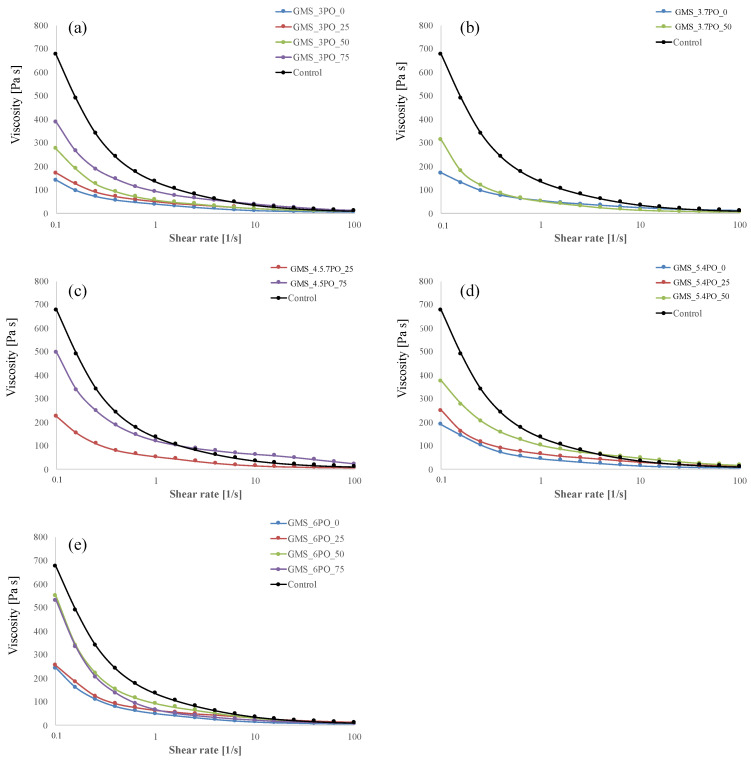
Apparent viscosity as a function of shear rate for all investigated cocoa hazelnut spreads: (**a**) GMS 3% OO oleogel with d ifferent PO substitution levels; (**b**) GMS 3.7% OO oleogel with different PO substitution levels; (**c**) GMS 4.5% OO oleogel with different PO substitution levels; (**d**) GMS 5.4% OO oleogel with different PO substitution levels; (**e**) GMS 6% OO oleogel with different PO substitution levels.

**Figure 2 foods-12-03122-f002:**
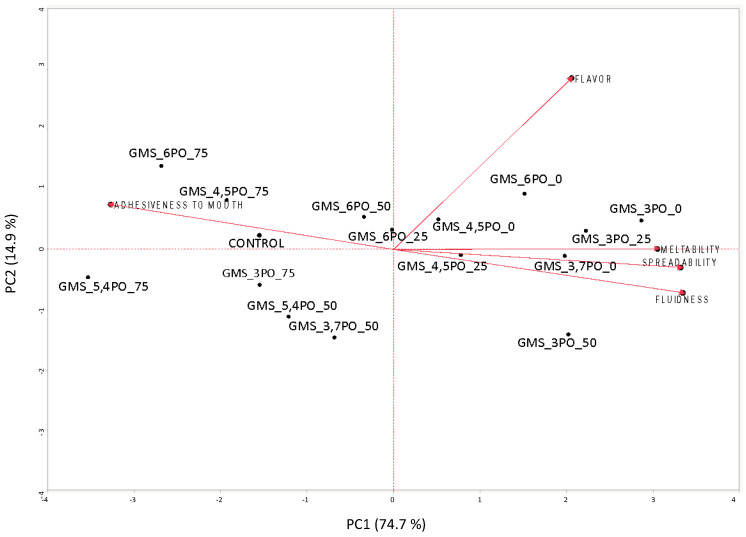
PCA Bi-plot (score and loadings) of descriptive sensory attributes of cocoa hazelnut spread samples.

**Table 1 foods-12-03122-t001:** Meaning of sensory attributes.

Parameters	Attribute Category	Evaluation
Fluidness	Non-oral texture	Dip the spoon in the cup and evaluate the rapidity of detachment
Spreadability	Non-oral texture	Evaluate the ease with which the sample is spread. Use the biscuit and the knife to perform the test.
Meltability	Oral texture	Place the sample between the tongue and palate and evaluate its melting rate
Adhesiveness to mouth	Oral texture	After meltability assessment, evaluate the force required to remove the sample from the palate using the tongue
Flavour	Gustative	Taste the cream and evaluate the intensity of the overall flavour

**Table 2 foods-12-03122-t002:** D-Optimal Design applied to optimize the cocoa hazelnut spread formulation.

	Compositions			
	Oleogel [%]		Fat Mixture [%]	
Formulation	x_1_	x_2_	z_1_	z_2_
GMS_3PO_0	97.00	3.00	100.00	0.00
GMS_3PO_25	97.00	3.00	75.00	25.00
GMS_3PO_50	97.00	3.00	50.00	50.00
GMS_3PO_75	97.00	3.00	25.00	75.00
GMS_3.7PO_0	96.00	3.70	100.00	0.00
GMS_3.7PO_50	96.30	3.70	50.00	50.00
GMS_4.5PO_25	95.50	4.50	75.00	25.00
GMS_4.5PO_75	95.50	4.50	25.00	75.00
GMS_5.4PO_0	94.60	5.40	100.00	0.00
GMS_5.4PO_50	94.60	5.40	50.00	50.00
GMS_5.4PO_75	94.60	5.40	25.00	75.00
GMS_6PO_0	94.00	6.00	100.00	0.00
GMS_6PO_25	94.00	6.00	75.00	25.00
GMS_6PO_50	94.00	6.00	50.00	50.00
GMS_6PO_75	94.00	6.00	25.00	75.00

**Table 3 foods-12-03122-t003:** Experimental data for each response (Casson’s viscosity, spreadability, and OBC) corresponding to a given set of spread compositions.

Formulation	*η_Ca_* [Pa s]	Spreadability [N mm]	OBC [%]
	y_1_	y_2_	y_3_
GMS_3PO_0	3.12 ± 0.50 ^a^	8.47 ± 0.11 ^a^	93.03 ± 0.21 ^a^
GMS_3PO_25	6.22 ± 0.35 ^d^	22.15 ± 1.81 ^b^	99.12 ± 0.11 ^d^
GMS_3PO_50	7.45 ± 0.72 ^e^	24.80 ± 0.08 ^b^	99.19 ± 0.26 ^d^
GMS_3PO_75	9.01 ± 1.01 ^f^	43.63 ± 1.29 ^e^	99.59 ± 0.23 ^d^
GMS_3.7PO_0	4.46 ± 0.23 ^b^	9.31 ± 1.48 ^a^	94.90 ± 0.74 ^b^
GMS_3.7PO_50	7.86 ± 0.45 ^e^	25.06 ± 0.33 ^b^	98.56 ± 0.54 ^c^
GMS_4.5PO_25	9.05 ± 0.89 ^f^	23.01 ± 2.75 ^b^	98.27 ± 0.39 ^c^
GMS_4.5PO_75	10.34 ± 1.00 ^f^	37.80 ± 1.96 ^d^	97.77 ± 0.57 ^c^
GMS_5.4PO_0	5.29 ± 0.34 ^c^	10.88 ± 0.49 ^a^	96.95 ± 0.73 ^c^
GMS_5.4PO_50	9.13 ± 0.41 ^f^	32.02 ± 0.80 ^c^	97.89 ± 0.18 ^c^
GMS_5.4PO_75	14.66 ± 1.11 ^g^	28.53 ± 3.12 ^b^	99.97 ± 0.01 ^d^
GMS_6PO_0	7.46 ± 0.25 ^e^	10.23 ± 1.07 ^a^	98.93 ± 0.23 ^c^
GMS_6PO_25	9.48 ± 0.41 ^f^	28.81 ± 1.60 ^b^	97.04 ± 0.06 ^c^
GMS_6PO_50	10.31 ± 0.67 ^f^	32.71 ± 1.66 ^c^	98.79 ± 0.17 ^c^
GMS_6PO_75	21.52 ± 0.87 ^h^	58.14 ± 0.02 ^f^	99.85 ± 0.04 ^d^

Different letters (a–h) in the same column reveal significant differences (*p* < 0.05) between the samples.

**Table 4 foods-12-03122-t004:** Statistical parameters and estimated parameters for independent variables in the prediction models for each response.

	*η_Ca_*	Spreadability	OBC
Model	Significant	Significant	Significant
R^2^	0.9270	0.9475	0.9316
Adjusted R^2^	0.7956	0.8529	0.8084
*p* value	0.0223	0.0103	0.0192
F value	7.0538	10.0217	7.5632
Lack of fit	Not significant	Not significant	Not significant
Standard deviation	1.6560	6.4583	0.8603

**Table 5 foods-12-03122-t005:** Best regression parameters identified for Casson’s viscosity (*η_ca_*), spreadability, and oil binding capacity (OBC).

	*η_ca_*		Spreadability		OBC	
Regression Parameter	*i* = 1	*p* Value	*i* = 2	*p* Value	*i* = 3	*p* Value
$\beta_{i1}$	**6.4577**	0.0007	**24.6352**	0.0091	**97.7534**	<0.0001
$\beta_{i2}$	**12.1929**	<0.0001	**36.3712**	0.0017	**98.6873**	<0.0001
$\beta_{i12}$	−6.0255	0.2432	−37.2292	0.2878	−3.4975	0.2248
$\beta_{i13}$(*z*_2_ = 0%)	−3.0657	0.0583	−14.8467	0.1455	**−4.4319**	0.0014
$\beta_{i13}$(*z*_2_ = 25%)	0.6647	0.6466	−0.2061	0.9833	1.7214	0.0718
$\beta_{i13}$(*z*_2_ = 50%)	1.0190	0.4527	0.6260	0.9448	1.1973	0.1452
$\beta_{i13}$(*z*_2_ = 75%)	1.3821	0.3581	14.4269	0.1455	1.5133	0.1022
$\beta_{i23}$(*z*_2_ = 0)	**−4.4681**	0.0138	−20.4994	0.0558	0.3958	0.5785
$\beta_{i23}$(*z*_2_ = 25%)	−1.8108	0.2405	−5.2821	0.5964	−1.2925	0.1472
$\beta_{i23}$(*z*_2_ = 50%)	−1.3212	0.3228	1.9758	0.8208	−0.5767	0.4275
$\beta_{i23}$(*z*_2_ = 75%)	**7.6002**	0.0013	**23.8057**	**0.0319**	1.4733	0.0732
$\beta_{i123}$	8.8654	0.3876	51.1299	0.4632	5.1963	0.3545

**Table 6 foods-12-03122-t006:** Casson’s parameters of cocoa hazelnut spread samples.

Sample	Casson’s Plastic Viscosity *η_Ca_* [Pa∙s]	Casson’s Yield Stress *τ_Ca_* [Pa]	R^2^
GMS_3PO_0	3.12 ± 0.50 ^a^	39.67 ± 1.21 ^d^	0.98
GMS_3PO_25	6.22 ± 0.35 ^d^	38.76 ± 1.11 ^d^	0.97
GMS_3PO_50	7.45 ± 0.72 ^e^	33.23 ± 2.12 ^c^	0.98
GMS_3PO_75	9.01 ± 1.01 ^f^	30.35 ± 1.01 ^c^	0.99
GMS_3.7PO_0	4.46 ± 0.23 ^b^	23.97 ± 1.25 ^b^	0.99
GMS_3.7PO_50	7.86 ± 0.45 ^e^	54.82 ± 2.28 ^e^	0.98
GMS_4.5PO_25	9.05 ± 0.89 ^f^	50.18 ± 2.12 ^e^	0.99
GMS_4.5PO_75	10.34 ± 1.00 ^f^	52.61 ± 2.78 ^e^	0.96
GMS_5.4PO_0	5.29 ± 0.34 ^c^	14.88 ± 1.56 ^a^	0.99
GMS_5.4PO_25	9.13 ± 0.41 ^f^	21.67 ± 1.99 ^b^	0.99
GMS_5.4PO_50	14.66 ± 1.11 ^g^	48.99 ± 1.21 ^e^	0.97
GMS_6PO_0	7.46 ± 0.25 ^e^	25.78 ± 1.02 ^b^	0.99
GMS_6PO_25	9.48 ± 0.41 ^f^	26.83 ± 1.06 ^b^	0.98
GMS_6PO_50	10.31 ± 0.67 ^f^	68.28 ± 2.04 ^f^	0.99
GMS_6PO_75	21.52 ± 0.87 ^h^	48.35 ± 2.13 ^e^	0.98
Control	6.77 ± 0.23 ^d^	85.19 ± 2.34 ^g^	0.97

Different letters (a–h) in the same column reveal significant differences (*p* < 0.05) between the samples.

**Table 7 foods-12-03122-t007:** Oil binding capacity of cocoa hazelnut spread samples.

Sample	OBC [%]
GMS_3PO_0	93.03 ± 0.21 ^a^
GMS_3PO_25	99.12 ± 0.11 ^d^
GMS_3PO_50	99.19 ± 0.26 ^d^
GMS_3PO_75	99.59 ± 0.23 ^d^
GMS_3.7PO_0	94.90 ± 0.74 ^b^
GMS_3.7PO_50	98.56 ± 0.54 ^c^
GMS_4.5PO_25	98.27 ± 0.39 ^c^
GMS_4.5PO_75	97.77 ± 0.57 ^c^
GMS_5.4PO_0	96.95 ± 0.73 ^c^
GMS_5.4PO_25	97.89 ± 0.18 ^c^
GMS_5.4PO_50	99.97 ± 0.01 ^d^
GMS_6PO_0	98.93 ± 0.23 ^c^
GMS_6PO_25	97.04 ± 0.06 ^c^
GMS_6PO_50	98.79 ± 0.17 ^c^
GMS_6PO_75	99.85 ± 0.04 ^d^
Control	99.95 ± 0.03 ^d^

Different letters (a–d), in the same column reveal significant differences (*p* < 0.05) between the samples.

**Table 8 foods-12-03122-t008:** Hardness and spreadability of cocoa hazelnut spread samples.

Sample	Hardness [N]	Spreadability [N mm]
GMS_3PO_0	2.84 ± 0.11 ^a^	8.47 ± 0.11 ^a^
GMS_3PO_25	6.65 ± 0.43 ^b^	22.15 ± 1.81 ^b^
GMS_3PO_50	7.13 ± 0.14 ^b^	24.80 ± 0.08 ^b^
GMS_3PO_75	12.78 ± 0.28 ^c^	43.63 ± 1.29 ^e^
GMS_3.7PO_0	3.03 ± 1.22 ^a^	9.31 ± 1.48 ^a^
GMS_3.7PO_50	6.34 ± 0.26 ^b^	25.06 ± 0.33 ^b^
GMS_4.5PO_25	7.28 ± 1.20 ^b^	23.01 ± 2.75 ^b^
GMS_4.5PO_75	11.49 ± 1.32 ^c^	37.80 ± 1.96 ^d^
GMS_5.4PO_0	2.70 ± 0.21 ^a^	10.88 ± 0.49 ^a^
GMS_5.4PO_25	8.86 ± 0.58 ^b^	32.02 ± 0.80 ^c^
GMS_5.4PO_50	8.35 ± 0.84 ^b^	28.53 ± 3.12 ^b^
GMS_6PO_0	3.50 ± 0.27 ^a^	10.23 ± 1.07 ^a^
GMS_6PO_25	8.48 ± 0.10 ^b^	28.81 ± 1.60 ^b^
GMS_6PO_50	8.13 ± 1.58 ^b^	32.71 ± 1.66 ^c^
GMS_6PO_75	32.12 ± 0.92 ^e^	58.14 ± 0.02 ^f^
Control	16.92 ± 0.53 ^d^	61.94 ± 2.96 ^f^

Different letters (a–f), in the same column reveal significant differences (*p* < 0.05) between the samples.

## Data Availability

The data presented in this study are available on request from the corresponding author.

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
