# Peer review of "Optimization of Hazelnut Spread Based on Total or Partial Substitution of Palm Oil"

_foods, 2023, doi:10.3390/foods12163122_

Round 1
Reviewer 1 Report
General comments:
The manuscript entitled “Optimization of hazelnut spread based on total or partial substitution of palm oil” present some valuable results on palm oil replacers using oleogel formulations. Hoewever, before a possible data publication some points need to be clarified.
All the text must be formated.
Specific comments:
1. The “viscosisity” term to indicate the rheological behavior of non-newtonian fluid is used in the many part of the text. The correct term is “apparent viscosity”.
2. The temperature of 3.2 °C was presented as the palm oil melting point. This values is very low for thi oil type.
3. In Section 2.2 the authors reports a oleogel cooling rate of 2.°C/ min. This process was carried out under agitation ?
4. The authors must to present the references from literature for the experimental methodologies used for texture measuremts and oil binding capacities, sections 2.3.1 and 2.3.2.
5. The statement “Casson viscosity increases as increase GMS content in the oleogels as well as palm oil content indicating higher necessary force to flow the sample due to a high strength structure of the fat phase.” is not completely correct.
The data in Table 6 do no present a clear tendency on apparent viscosity increasing with GMS increasing.
6. The use of bar graphs to represent OBC, hardness and spreadbility it was not a good choice. The autors must to try other graphic type or even the use of Tables.
7. The statement “Since no differences in the process parameters employed for the refining of all cocoa hazelnut spread samples, as well as in the formulation of ingredients in powder (sugars, cocoa), the constant particle size distribution has been assumed.” is not completely true. “
During the production of hazelnut spread based all the ingredients were solubilizied and recristalized. The particle size distribution was resulto f the cooling process and system composition.
8. The melting point of the systems is an important information to discuss the physical properties and the sensory results. If the authors have those data, they must to report in this manuscript and correlate them with the other resuts reported.
9. The authors presented the statement “Kangchai et al. [27] reported that hardness is significantly associated with spreadability, the higher the hardness, the greater the resistance to spread (less spreadability).” From this point of view, hardness and spreadbility show inverse behavior, i.e. the increase of the hardness result on the decrease of the spreadability.
However, this behavior was not observed in the graph bars or along the discussion.
A full english language review should be done.
Author Response
Answer to Reviewer 1
Manuscript ID: foods-2552255: Optimization of hazelnut spread based on total or partial substitution
of palm oil.
We gratefully acknowledge the Reviewer for his thorough and careful examination of the manuscript. We have considered his comments/suggestions and revised the paper accordingly. Additions and changes in the manuscript are marked in red.
Reviewer
The manuscript entitled “Optimization of hazelnut spread based on total or partial substitution of palm oil” present some valuable results on palm oil replacers using oleogel formulations. However, before a possible data publication some points need to be clarified.
All the text must be formatted.
The text has been formatted according to guidelines for authors.
Specific comments:
- The “viscosity” term to indicate the rheological behavior of non-newtonian fluid is used in the many part of the text. The correct term is “apparent viscosity”.
The term viscosity has been replaced with “apparent viscosity”. See the revised manuscript.
- The temperature of 3.2 °C was presented as the palm oil melting point. This values is very low for thi oil type.
Sorry but in the section “2.1 Materials” of the manuscript there was a typo. The correct melting temperature is 32.34°C.
In our previous studies, the heating thermogram of crude palm oil (figure reported below) revealed melting temperatures of 21.10°C and 32.34°C close to 19.45°C and 34.48°C found by Che Man et al., 1999 (Y.B. Che Mana, T. Haryatia, H.M. Ghazalib, and B.A. Asbia. Composition and Thermal Profile of Crude Palm Oil and Its Products, Journal of the American oil chemists' society, 1999).
- In Section 2.2 the authors reports a oleogel cooling rate of 2.°C/ min. This process was carried out under agitation?
The process has been carried out under static conditions. The information has been added to the section “2.2 Oleogel preparation” of the revised manuscript.
- The authors must to present the references from literature for the experimental methodologies used for texture measurements and oil binding capacities, sections 2.3.1 and 2.3.2.
Texture measurements have been carried out according to Kangchai et al 2018, while oil binding capacity was evaluated, as already reported in the manuscript, according to the procedure of Aydemir et al. 2019. The new reference has been added to the revised manuscript. See the revised manuscript.
- The statement “Casson viscosity increases as increase GMS content in the oleogels as well as palm oil content indicating higher necessary force to flow the sample due to a high strength structure of the fat phase.” is not completely correct. The data in Table 6 do no present a clear tendency on apparent viscosity increasing with GMS increasing.
The revised manuscript has been modified as follows: “Given a certain composition of GMS in the oleogel, the Casson viscosity increases as the palm oil content in the fat phase increases indicating that a higher force is necessary to let the sample flowing, because of the high strength structure of the fat phase. Without palm oil in the fat phase, the Casson viscosity significantly increases when the composition of GMS in the oleogel increases too. With palm oil composition in the fat phase counts 25%, or 50% or 75% the role of GMS composition in determining the apparent viscosity of the spread cannot be generalized. For palm oil composition of 25%, the apparent viscosity significantly increases when GSM % in the oleogel moves from 3% to 4.5%. Further increasing in GSM composition does not lead to any appreciable increment of the apparent viscosity.”
- The use of bar graphs to represent OBC, hardness and spreadability it was not a good choice. The authors must to try other graphic type or even the use of Tables.
According to the reviewer’s suggestion, OBC, hardness and spreadability graphs have been replaced with tables. See the revised manuscript.
- The statement “Since no differences in the process parameters employed for the refining of all cocoa hazelnut spread samples, as well as in the formulation of ingredients in powder (sugars, cocoa), the constant particle size distribution has been assumed.” is not completely true.“ During the production of hazelnut spread based all the ingredients were solubilizied and recristalized. The particle size distribution was result of the cooling process and system composition.
According to the reviewer’s suggestions, the revised manuscript has been modified as follows: ” For this latter, no significant differences were found about the particle size distribution for all the spread samples, which ranged from 26.30±0.35 μm to 33.48±0.76 μm.”
- The melting point of the systems is an important information to discuss the physical properties and the sensory results. If the authors have those data, they must to report in this manuscript and correlate them with the other resuts reported.
Dear reviewer, we haven’t these data because the principal aim of this study was to investigate the possibility of partial and total replacement of palm oil with an olive oil-based oleogel in spread formulation by studying the main quality parameters of cocoa hazelnut spreads.
- The authors presented the statement “Kangchai et al. [27] reported that hardness is significantly associated with spreadability, the higher the hardness, the greater the resistance to spread (less spreadability).” From this point of view, hardness and spreadbility show inverse behavior, i.e. the increase of the hardness result on the decrease of the spreadability. However, this behavior was not observed in the graph bars or along the discussion.
As reported by different authors (Kangchai et al. 2018; Abdolmaleki et al., 2022), spreadability is defined as the resistance to spread. From this point of view, hardness and spreadability have the same behaviour: the increase of hardness results in the increase of resistance to spread (increase of spreadability value). For a better understanding, the revised manuscript has been modified as follows: “The hardness and spreadability of the different cocoa hazelnut spreads were reported in table 8. Our results were in agreement with several authors [23, 28] who reported that hardness is significantly associated with spreadability, the higher the hardness, the greater the resistance to spread.”

Reviewer 2 Report
The introduction is complete and well structured, and include all relevant references . Material and methods are in general described . The results are well-written, and the discussion is appropriate. The conclusion section is too long. The sensory evaluation has not been described: How were the panelists trained? The sensory trials were conducted by respecting the ethical code? The 9 panelists used in total are a few number
Minor editing of English language required
Author Response
Answer to Reviewer 2
Manuscript ID: foods-2552255: Optimization of hazelnut spread based on total or partial substitution
of palm oil.
We gratefully acknowledge the Reviewer for his thorough and careful examination of the manuscript. We have considered his comments/suggestions and revised the paper accordingly. Additions and changes in the manuscript are marked in red.
Reviewer
The introduction is complete and well structured, and include all relevant references . Material and methods are in general described . The results are well-written, and the discussion is appropriate. The conclusion section is too long. The sensory evaluation has not been described: How were the panelists trained? The sensory trials were conducted by respecting the ethical code? The 9 panelists used in total are a few number
According to the reviewer’s suggestion the section “2.3.4 Sensory analysis” of the manuscript has been modified as follows: “The judges were chosen among members of the Department of Industrial Engineering of the University of Salerno using as selection criteria time availability, no nuts allergy/intolerance. The preselected judges were selected on their ability to identify odours and the five basic tastes, using ISO 3972, ISO 5496, to recruit 9 judges. The descriptors chosen for the sensory analysis of spread samples, as well as their evaluation method, are reported in table 1. All procedures performed in this study involving human participants were in accordance with the 1964 Helsinki Declaration and its later amendments”

Reviewer 3 Report
Hi dear Editorial board and the respected authors
This article "Optimization of hazelnut spread based on total or partial substitution of palm oil” was revised and has a novelty and I recommend it for publication after consideration of the following comments.
Title: If you can rewrite and make it more interesting for readers. I propose: “Prosopis alba seed as a functional food waste for food formulation enrichment”.
Abstract:
· Glycerol monostearate/olive oil-based oleogel. What is the influence of olive oil?
· Design of Experiment (DoE) is it right? I think is wrong. Design of expert seems right.
· The type of statistical design used in this research should be mentioned as a detail i.e., factorial and axial repeat? α? Dependent and independent variables etc?
· Please express the base of selecting the optimizing formula for your study?
Introduction:
· Please express about the RSM in a paragraph and try to cite the following new references in the throughout the manuscript text: Journal of food processing and preservation 45 (4), e15311.
Materials:
· Please write materials as Company Name (City, Country), especially for chemical analysis assessment which used in the study.
Methodology:
· The way of expressing the method of measuring macronutrients and other parameters has a scientific flaw. Please take help from the following article for the correct way of expressing it, so that the standard number of the working method should be clearly stated (https://doi.org/10.1590/fst.60820).
· Confusion in metric units is seen throughout the text.
· Table 2: please explain about the x and z in the table and please provide each tables or figures as a self-explanatory approach.
“Results:
· All Tables: The alphabetical statistical letters for the means should all be modified such that the greatest number has the letter a and as the numbers go lower, letters b, c etc.
· Table 6: Please express the title of the table as a completely heading and prevent concise titles.
· Fig. 2: Please compare the statistical analysis as alphabetic letters inserting the above the columns.
Discussion:
Discussion text must grammar improve and in some cases it is very weak and maybe there is no discussion at all.
Conclusions:
Conclusion is very general, try to make it more scientific, comprehensive and concise in detail, especially.
References: It is OK.
The article has many flaws in express and concept of English, it is suggested to be revised in a scientific and native way.

The article has many flaws in express and concept of English, it is suggested to be revised in a scientific and native way.
Author Response
Answer to Reviewer 3
Manuscript ID: foods-2552255: Optimization of hazelnut spread based on total or partial substitution
of palm oil.
We gratefully acknowledge the Reviewer for his thorough and careful examination of the manuscript. We have considered his comments/suggestions and revised the paper accordingly. Additions and changes in the manuscript are marked in red.
Reviewer
Hi dear Editorial board and the respected authors
This article "Optimization of hazelnut spread based on total or partial substitution of palm oil” was revised and has a novelty and I recommend it for publication after consideration of the following comments.
Title: If you can rewrite and make it more interesting for readers. I propose: “Prosopis alba seed as a functional food waste for food formulation enrichment”.
The authors believe that the reviewer is writing about some other manuscript under revision since the proposed title has nothing to do with the content of our manuscript. No action will be taken about this comment.
Abstract:
- Glycerol monostearate/olive oil-based oleogel. What is the influence of olive oil?
As reported in the Introduction, oleogels are solid-like fats obtained from liquid vegetable oil and specific substances, termed oleogelators, able to create a three-dimensional network that entraps the liquid oil forming an oleogel. olive oil is the oil used for the preparation of oleogel and one of the aims of our study has been to evaluate the amount of GMS to add to the olive oil for the preparation of an oleogel suitable for the total replacement of palm oil for the manufacturing of hazelnut spread.
Design of Experiment (DoE) is it right? I think is wrong. Design of expert seems right.
With acronyms DoE we meant Design of Experiments. Dozens of papers dealing with this topic are available in the literature, not just about food tech.
The type of statistical design used in this research should be mentioned as a detail i.e., factorial and axial repeat? α? Dependent and independent variables etc?
The abstract has been modified according to the reviewer’s suggestion.
Please express the base of selecting the optimizing formula for your study?
According to the reviewer’s suggestion, we modify the sentence as follows: “In this study, the desirability function corresponded to an optimal formulation set by minimizing Casson’s viscosity and spreadability, while maximizing the oil binding capacity, in a range of values referred to a market leader cocoa hazelnut spread”
Introduction:
- Please express about the RSM in a paragraph and try to cite the following new references in the throughout the manuscript text: Journal of food processing and preservation 45 (4), e15311.
According to the reviewer’s suggestion, a new reference (Mirani and Goli, 2020) has been added to the revised manuscript.
Materials:
- Please write materials as Company Name (City, Country), especially for chemical analysis assessment which used in the study.
Done
Methodology:
- The way of expressing the method of measuring macronutrients and other parameters has a scientific flaw. Please take help from the following article for the correct way of expressing it, so that the standard number of the working method should be clearly stated (https://doi.org/10.1590/fst.60820).
This comment is probably referring to some other manuscript.
- Confusion in metric units is seen throughout the text.
All units used in the manuscript were checked and corrected accordingly. If there is any specific entity for which the units are wrong, the reviewer is kindly requested to provide a detailed comment on it.
- Table 2: please explain about the x and z in the table and please provide each tables or figures as a self-explanatory approach.
The meaning of x and z in table 2 is given in section 2.4
“Results:
- All Tables: The alphabetical statistical letters for the means should all be modified such that the greatest number has the letter a and as the numbers go lower, letters b, c
- Table 6: Please express the title of the table as a completely heading and prevent concise titles.
- 2: Please compare the statistical analysis as alphabetic letters inserting the above the columns.
All required and suggested actions were taken by the authors when preparing the new version of the manuscript
Discussion:
Discussion text must grammar improve and in some cases it is very weak and maybe there is no discussion at all.
The discussion has been enriched. The English writing style will be checked by the English-proof service of Foods.
Conclusions:
Conclusion is very general, try to make it more scientific, comprehensive and concise in detail, especially.
Conclusion was improved and enriched
References: It is OK.
The article has many flaws in express and concept of English, it is suggested to be revised in a scientific and native way.
The English writing style will be checked by the English-proof service of Foods.

Reviewer 4 Report
The article presents an interesting use of oleogels to replace palm oil in a hazelnut spread.
The experiment described in the article was correctly planned using proper analytical methodology.
The only concern is the lack of information on the bioethics committee's approval of the sensory study, or at least information on the formal consent given by the panelists, which informed of all possible side effects of the analyzed product (nut allergy, MSG consumption).
Please complete this information.
Author Response
Answer to Reviewer 4
Manuscript ID: foods-2552255: Optimization of hazelnut spread based on total or partial substitution
of palm oil.
We gratefully acknowledge the Reviewer for his thorough and careful examination of the manuscript. We have considered his comments/suggestions and revised the paper accordingly. Additions and changes in the manuscript are marked in red.
Reviewer
The article presents an interesting use of oleogels to replace palm oil in a hazelnut spread.
The experiment described in the article was correctly planned using proper analytical methodology.
The only concern is the lack of information on the bioethics committee's approval of the sensory study, or at least information on the formal consent given by the panelists, which informed of all possible side effects of the analyzed product (nut allergy, MSG consumption).
Please complete this information.
This study was conducted according to the guidelines of the Declaration of Helsinki.
Participants gave informed consent before participating in this study, about which appropriate information was provided regarding nuts allergy/intolerance. This information has been added to the revised manuscript.

Round 2
Reviewer 1 Report
The manuscript is suitable for publication.
Author Response
Thank you.
Reviewer 3 Report
The statistical analysis in tables especially was not synchronized especially for Table 3 for OBC factor
Tables were not self-explanatory
· All Tables: The alphabetical statistical letters for the means should all be modified such that the greatest number has the letter a and as the numbers go lower, letters b, c etc.
· Conclusion is very general, try to make it more scientific, comprehensive and concise in detail, especially.
Author Response
Answer to Reviewer 3
Manuscript ID: foods-2552255: Optimization of hazelnut spread based on total or partial substitution
of palm oil.
We gratefully acknowledge the Reviewer for his thorough and careful examination of the manuscript. We have considered his comments/suggestions and revised the paper accordingly. Additions and changes in the manuscript are marked in red.
Reviewer
The statistical analysis in tables especially was not synchronized especially for Table 3 for OBC factor
Tables were not self-explanatory
All Tables: The alphabetical statistical letters for the means should all be modified such that the greatest number has the letter a and as the numbers go lower, letters b, c etc.
All suggested actions were taken by the authors in the revised manuscript
Conclusion is very general, try to make it more scientific, comprehensive and concise in detail, especially
Conclusion section has been improved

Reviewer 4 Report
The authors have introduced all the necessary changes. The manuscript is ready for publication.
Author Response
Thank you.